# An Optimizing Heat Consumption System Based on BMS

Nicoleta Cristina Gaitan [1,2,]*, Ioan Ungurean [1,2,]*, Costica Roman [3,4,]* and Catalin Francu [5,]*

1   Faculty of Electrical Engineering and Computer Science, Stefan cel Mare University of Suceava,
    720229 Suceava, Romania
2   MANSiD Integrated Center, Stefan cel Mare University, 720229 Suceava, Romania
3   Faculty of Economics, Administration and Business, Stefan cel Mare University, 720229 Suceava, Romania
4   S.C.CAOM S.A, 705200 Pascani, Romania
5   SC Building Technology Group R SRL, 700202 Iasi, Romania
*   Correspondence: cristinag@eed.usv.ro (N.C.G.); ioanu@eed.usv.ro (I.U.); romanc@seap.usv.ro (C.R.);
    catalin.francu@btgromania.ro (C.F.)

**Abstract:** The recent trend is to transform existing buildings into smart, green, or passive buildings by adopting sustainable technologies to achieve increased life comfort and production efficiency through reducing carbon emissions, energy consumption, and operating costs. Thus, existing buildings can be rehabilitated in order to have the lowest possible energy consumption. One of the highest demands on the energy consumption of buildings is the heating system used during the winter months. In this paper, we propose a management and control system for electrical and energy heating consumption, based on a BMS (building management system) that is used for the rehabilitation of the students' dormitories on a university campus. This includes a cogeneration plant that can be controlled in order to produce electrical and heating energy, depending on the requirements needed to heat the building in the cold season. The system reduced the consumption of energy for heating by 13% and of electricity by 32%.

**Keywords:** smart system; smart building; energy efficiency; building management system





## 1. Introduction

In traditional heating and cooling systems, system management is performed through so-called heating/cooling sources, which are usually a heating radiator or an air-conditioning unit, and cold or hot water, which is distributed around the building without any control [1].

In most cases, room radiators are equipped with a manual valve, or, in more modern scenarios, the basic adjustment is made through a thermostatic valve. In any case, the room thermostats cannot be controlled by the systems for producing heat (central heating) or cold (air-conditioning system); thus, there is no control over the amount of energy consumed to heat/cool a room. For example, a window or door may be opened and then left open, but the room may still be being heated or cooled.

The use of a BMS (building management system) will allow engineers to acquire information from the environment through sensors (contactors closing/opening, temperature probes, etc.) and to interconnect both heating/cooling sources and the events that may occur in a room (temperature rises, windows being left open, etc.).

Based on these possibilities, the BMS system gathers all the necessary information to make an optimal decision regarding starting/stopping the heating/cooling system, in order to obtain maximum efficiency in that environment in terms of temperature and energy consumption. The BMS will open/close all systems at the right time to always ensure an optimum temperature in the managed premises.

The main objective of this paper is to propose a control system, based on BMS, that optimizes energy consumption to ensure the medium- and long-term functionality of the accommodation services of a campus. This is conducted for two indispensable and related services: the supply of electricity and of heating energy at a reduced price, in accordance

with current and future consumption and in agreement with European Community standards and directives on environmental protection. An energy rehabilitation solution based on the cogeneration of electricity and heat responds to Directive 2004/8/EC of the European Parliament and of the Council of 11 February 2004 [2] on the promotion of cogeneration, based on the demand for useful thermal energy in the domestic energy market.

In this paper, we propose a solution for a smart system based on BMS for the optimization of heating energy consumption. The scope of this system is to achieve the optimization of energy consumption in terms of the energy used to heat a building in the cold season. Optimization is performed via automatic control of the energy produced and supplied, depending on the requirements identified by the system. Thus, if a room is uninhabited, then the minimum possible temperature will be maintained in it, and if a room has an open window, the heating system in that room will be turned off, just to avoid wasting energy. This solution includes a cogeneration plant that produces electrical and thermal energy and this is used for the rehabilitation of the students' dormitories on a university campus. Thus, the contributions of this paper are the proposed architecture and the implementation of the management system.

This paper is structured as follows: Section 2 presents the background for the study and related work that highlights the advantages and benefits of BMS systems, along with the solution for a BMS presented in the specialized literature; Section 3 presents the general architecture of the proposed system and outlines the operation mode, the general SCADA (supervisory control and data acquisition) system for the cogeneration plant, and the main equipment needed should the solution be adopted. A discussion related to the proposed solution is presented in Section 4 and a case study and related discussions are presented in Section 5. Our final conclusions are presented in Section 6.

## 2. Background and Related Work

The BMS is an automation and control system [3] used in large buildings. These systems can control and monitor the systems that are specific to the buildings, such as lighting, power, security, heat/cold, and ventilation systems [4]. With the development of the concept of the IoT (Internet of Things), the BMS began to be integrated into this concept [4,5].

By using a BMS, numerous benefits can be obtained, from the simplest to the most sophisticated, without modifying the original heat/cold production systems. These benefits include:

- Management of thermal energy consumption by recording and adjusting the average temperature of the building; we can even determine and avoid potential system anomalies.
- Management of the minimum and maximum temperature thresholds for each room. Controlling the minimum–maximum thresholds will prevent any manual adjustment errors or malicious interference by the users. (For example, if the minimum-maximum threshold is set remotely from 10 °C to 23 °C by the system, no one can turn up or forget to turn off the radiator, so that the room temperature does not exceed 23 °C. The BMS system will turn it off automatically, regardless of the user's decision.
- Managing the temperature in each room according to circumstances, such as summer, winter, day, night, etc.
- Event management in each room, automatically adjusting the temperature when a door or window is open.

    All of these features will lead to the following objectives:

- A comfortable, managed environment;
- Energy savings;
- The smart control of energy consumption;
- The smart control of system usage.

Additional options can be added at any time after the system has been installed and configured:

- The heating/cooling control depends on the occupancy of a room. This can be determined using door contactors and/or IR detectors, or any other type of sensor.
- The management and control of the heating/cooling system by adding alternative energy sources, such as solar energy, cogeneration energy, etc., and optimizing their use.

Using a BMS system, if viewed in a more global way, we can integrate other "domains" of a building. Energy-saving, in general, can and must be applied to all areas of a building.

Previous experiences in hotels, nursing homes, and domestic dwellings have led us to the conclusion that the energy savings due to the installation of such a system are between 15% and 50%, depending on the location, the level of integration of the BMS, geographical location, etc.

In several recent, specialized articles, the authors have shown that, depending on the design objectives, ordinary buildings can evolve and transform into smart buildings. In [6] the authors propose the use of renewable energy sources for the smart management of home energy. The use of renewable sources for thermal load management is proposed in [7]. The use of a sensor network for structural health diagnosis is proposed in [8]. In [9], the authors propose a sensor network in order to diagnose problems when building HVAC systems. A solution for residential thermal energy management in terms of peak-load shifting is proposed in [10]. In [11], the authors present a review related to the sensing, controlling, and use of an IoT infrastructure in smart buildings. A solution based on a neural network's predictive control for energy management in zero-energy buildings is proposed in [12]. In [13] an autonomic management architecture for multi-HVAC systems is proposed. Artificial intelligence can be used for the management of energy consumption in smart buildings [14–16]. In the literature, different solutions for the optimization of energy consumption in smart buildings are proposed; some examples can be found in [17,18].

A smart token-based scheduling algorithm (smart-TBSA) system was developed to minimize energy use in HVAC (heating ventilation air-conditioning) systems in commercial construction and this was applied to a pilot building in Nanyang University of Technology, Singapore, as introduced and presented in [19]. Starting from an IoT prototype, consisting of hardware and software components and networks related to their integration, the authors solved problems such as the interaction between these components, as well as cloud-connectivity and the prototype's integration with legacy systems. They have demonstrated that the smart TBSA and IoT yield energy savings of up to 20% and can be transformed into a more adaptable system with intelligent, scalable, and decentralized control.

In [20], the authors presented an overview of the ongoing strategy for BEMS (building energy management systems) such as MPC (model predictive control), DSM (demand-side management), optimization, and FDD (fault detection and diagnosis) for improving energy efficiency. The authors also reviewed the subsystems and construction techniques used for each type of strategy, focusing, in particular, not only on the systems but also on the software used to validate the methodologies.

In the development of smart buildings, the authors of [21] proposed a real-time management system for controlling the various aspects of smart buildings, based on criteria for obtaining not only reliability and real-time adaptation to environmental conditions but also on machine learning over time in a real-life scenario, using the subjective parameters of the buildings in the control design, based on the model and on the optimization of the thermal or visual aspects of the buildings.

A system for an innovative nearly-zero energy building (nZEB) is proposed and presented in [22]. The proposed system is based on a prototype for a prefabricated nZEB. In [23], a strategy is proposed for the management of domestic water heaters, based on human behaviors and IoT concepts. This strategy has led to a decrease in energy consumption of between 20% and 34%. A smart IoT-based system that controls the air conditioner device is proposed in [24]. The indoor environment is controlled using sensors for the air temperature and relative humidity inside the room. In [25], an IoT-based plug-

and-play building thermal model learning framework was developed and investigated that is based on machine learning. The framework learned the building's thermal requirements and efficiency using energy resources.

## 3. The General System Architecture and the Operation Mode

This section presents the general architecture and operation of a heating control system for students' rooms or a dormitory. Within this system, in each room, there will be a temperature probe and an electro-valve to control the heating system. These are interconnected via a fieldbus (MODBUS TCP/IP) to enable communicating with them remotely. The solution also includes a cogeneration unit placed on-campus to produce heat; it has a connection to the central heating system used in the town, which can be used if the cogeneration unit does not cope well or it does not operate within optimal parameters (fail-tolerant).

The system will operate according to the following principles: each electro-valve and temperature probe will be connected to a PC in the general control room via a fieldbus (MODBUS TCP/IP). These will be operated and managed by a computer that will integrate this data, together with the data from the heat exchangers and the cogeneration group, obtaining asynchronous operations in accordance with the weather conditions and the restrictions imposed by users and administrators, the ultimate goal being to maximize the efficiency of the installation or, in other words, to save thermal energy.

Regarding the basic structure of the system, as well as its operation, we will follow Figure 1. In the building, there is a control room where a BMS is executed on an industrial computer. The BMS will send commands to both the transmission network and the thermal agent production system, to achieve optimal system operation and to prevent pressure and temperature issues in the system. In addition, there is a central system that will take over and centralize the data from the system. A SCADA system [26] that controls the cogeneration plant will also be installed on this central system. From the central computer, you can set the subscription time for each room. If a room is uninhabited, the ambient temperature will be set to the minimum value possible. If the sensors in the room detect that the window or door is left open, then the heating system will be switched off.

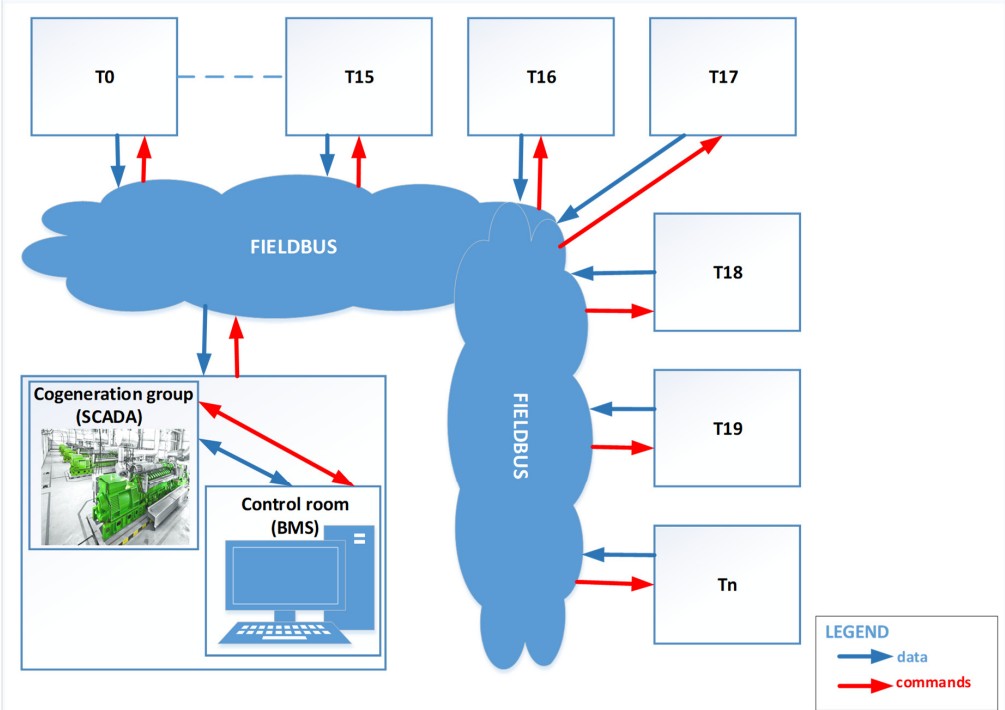

**Figure 1.** Data and command transmission architecture.

The data that is centralized in the system comes from both the external transport network, located outside the students' dormitories and, especially, from the internal distribution network located inside the building.

On the other hand, the transmitted commands will go to both the cogeneration group and the heat exchangers at the confluence of the external and internal networks within each home. All valves and electric pumps on each thermal agent column will also be electronically controlled (see Figure 2).

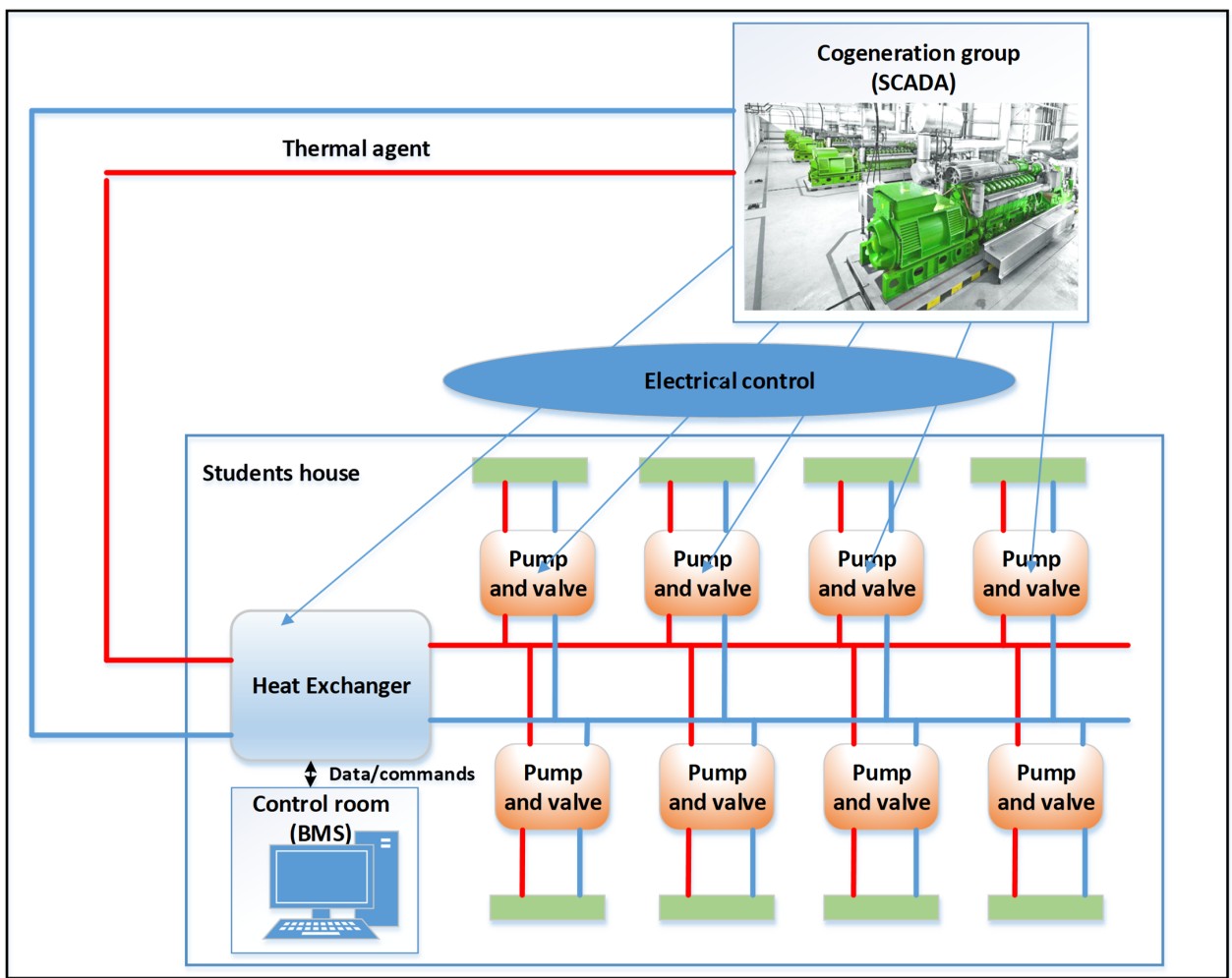

**Figure 2.** Cooling/heating agent from the building (students' dormitories).

When the temperature sensors indicate the desired parameters are reached, the solenoid valves recirculate the heating medium to the building level. If most of the solenoid valves are closed, then the heat exchanger shutdown command will be issued, thus stopping the heating agent from entering the building. Depending on this condition, the thermal energy production command will be transmitted to the cogeneration group, thus opting for coordination between the thermal energy production and the existing need in the system. This will lead to the achievement of the goal of optimizing consumption and streamlining the production and use of the thermal agent.

Based on the analysis, the information obtained led to the following concepts, namely:

- Thermal energy is provided through the thermal network of the campus, which uses heated water as the thermal agent, distributed through a network of metal or pexal pipes, and radiators are used as the heating source in the rooms.
- Heat control is performed at both building and room levels. The heating medium is transferred to the level of each room by means of a column that is connected to a

central heating system located in the basement of the buildings, which delivers heated water at 1 or 2 ends on each floor (see Figure 3).

- Each room will have a temperature probe installed, used to assess the ambient temperature—and for use as a landmark.
- The software system will coordinate the process of production, transport, and distribution of the thermal agent within the system.

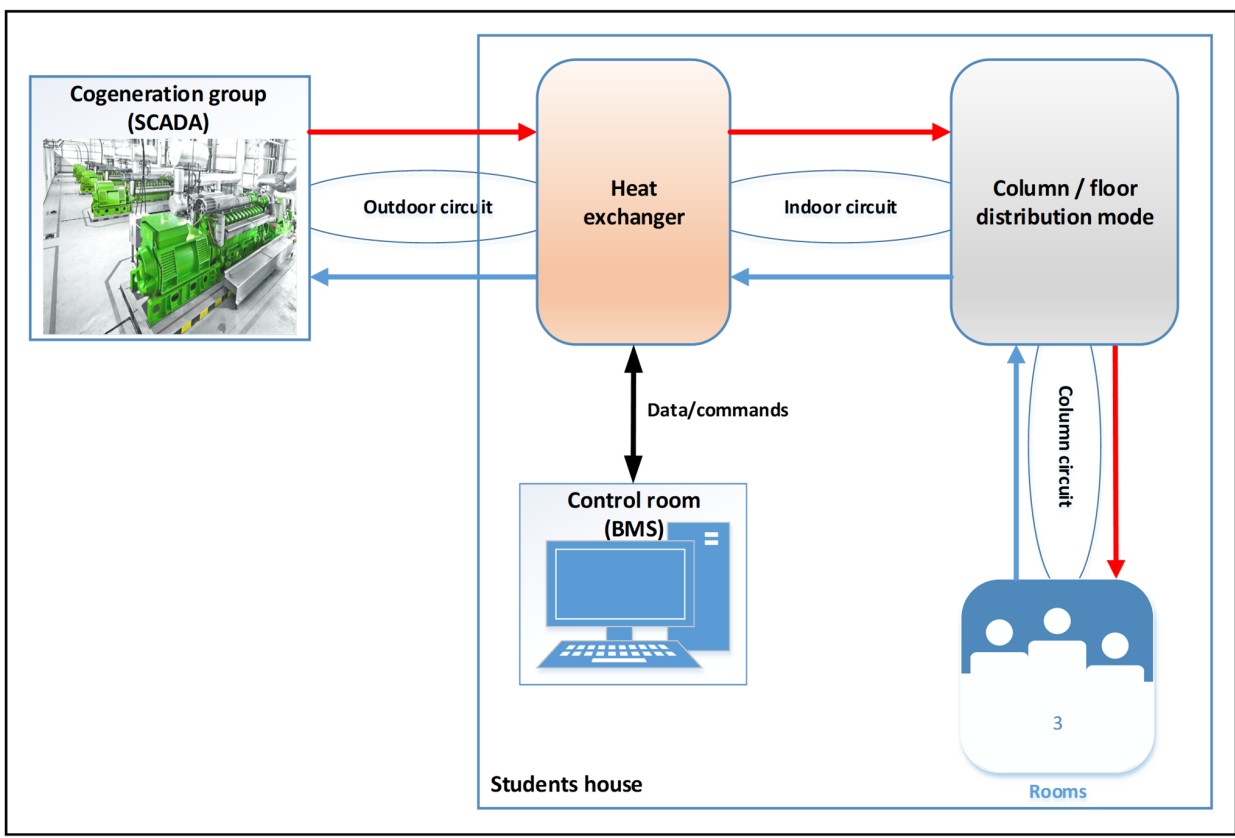

**Figure 3.** Circuit diagram of the existing thermal agent in the system.

The automation and control of a cogeneration group are achieved through a SCADA [27,28] software program, which in turn needs to communicate data and receive commands from the BMS.

## 4. The General SCADA System for the Cogeneration Plant

In order to create a decentralized management system to monitor, control, view, and record all the parameters of the cogeneration plant, as well as the creation of a database with these parameters and all the events and alarms from the equipment and machinery of the cogeneration plant, the building was provided with a SCADA system with a PLC, connected to a computer equipped with a touchscreen in an industrial configuration.

The control panel contains:

- The programmable controller that communicates on the MODBUS fieldbus with the main equipment in the plant, comprising motors and hot water boilers.
- I/O (input/output) blocks for interfacing with the process, respectively:
    - o  DI (digital input)—equipment status monitoring (pumps/shutdown operations, automatic valves (closed-open), etc.).
    - o  DO (digital outputs)—equipment commands, start/stop.
    - o  AO (analog output)—analog signal controls (transmission of prescribed values for the regulation of other equipment (e.g., a set-point transmission for generator power), automatic control valve controls, etc.).

o    AI (analog input)—unified input signals provided by temperature, pressure, transducers, etc.

- Uninterruptible power supply (UPS).

The automation elements/equipment from the cogeneration plant are connected to the operating station, respectively:

- Wired connections for unified voltage and current signals, and contact signals from:
  - o    Field automation equipment (pressure and temperature transducers, automatic valves);
  - o    the automation panels of the two gas thermal engines, the HWB (hot water boiler) and water-softening station panel;
  - o    General low-voltage switchboard.
- MODBUS network cable connections:
  - o    Automation panel for gas thermal engines;
  - o    A thermal energy calculation element for hot water, supplied by the two gas thermal engines and the primary district heating/cogeneration plant output circuit.
- Fieldbus cable connections with HWB automation panels;
- An Ethernet network, with the general dispatcher on campus and the BMS computer stations in the dormitories.

The main objective of this paper is to propose a control system based on BMS that optimizes energy consumption. The proposed system includes a system of automation control and supervision for primary thermal and electrical equipment, as well as auxiliary equipment. The main technical characteristics of the automation equipment will be listed later in the paper.

### 4.1. Technical Solution Adopted—Main Equipment

This section presents the main equipment used to implement the smart system proposal for optimizing heat consumption. The functional scheme of the solution that was adopted is presented in Figure 4.

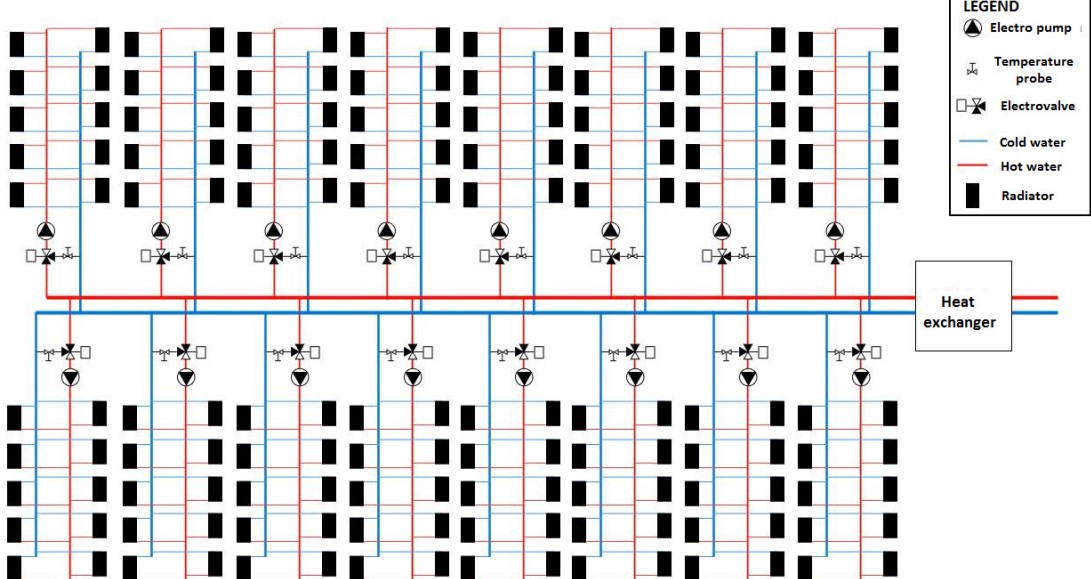

**Figure 4.** The functional scheme of the solution, adopted for a students' dormitory.

### 4.1.1. Cogeneration Plant

From the production point of view, the plant exhibits the following technological lines:

- Two lines of electricity and thermal energy production, with a cogeneration group containing:
  o An engine that runs on methane gas and directly drives the generator, to produce electricity at a voltage of 0.4 kV and 650 kW power.
  o An engine that runs on methane gas and directly drives the generator, to produce electricity at a voltage of 0.4 kV and 1300 kW power.
  o 4-stage heat recovery installation, respectively:
    - The first stage is a heat exchanger with water-to-water plates (intercooler stage 1), where a temperature increase of about 5 °C (from 70 °C to 75 °C and P = 300 kW) is achieved in the case of the large engine, and an increase of 3 °C (from 70 °C to 73 °C and P = 120 kW) in the case of the small motor.
    - The second stage is an oil-water heat exchanger (lubricating oil cooler), where a temperature increase of about 2 °C (from 75 °C to 77 °C and P = 140 kW) is achieved in the case of the large engine and of 2 °C (from 73 °C to 75 °C and P = 70 kW) in the case of a small engine.
    - The third stage is a water-to-water heat exchanger (engine-water cooling), where a temperature increase of about 7 °C (from 77 °C to 84 °C and P = 280 kW) is achieved in the case of the large engine and 5 °C (from 75 °C to 80 °C and P = 200 kW) in the case of a small engine.
    - The fourth stage is flue gas heat recuperation, where a temperature increase of about 8 °C (from 82 °C to 90 °C and P = 500 kW) was achieved in the case of the large engine, and 9 °C (from 81 °C to 90 °C and P = 330 kW) in the case of a small engine.
- Thermal energy production installation (hot water) is shown, with three hot water boilers.
  o The 3 hot water boilers, equipped with mixed natural gas-diesel burners, are:
    - Maximum thermal load = 2800 kW/pp;
    - T max = 105 °C;
    - Pn = 10 bar.
  o A liquid fuel supply installation (diesel), with a day's-capacity tank (V = 2000 L) for the three mixed burners, with flow and recirculation pipes for starting and running tests.

The operation of the thermomechanical scheme is economically based on the continuous operation of the two cogeneration groups that supply both electricity and thermal energy at an overall efficiency of 87%, with a limited thermal power of 2000 kW and parameters set at traducer/inverter = 90 °C/70 °C. When the demand for thermal energy increases, both due to the consumption of domestic hot water and due to the decrease in the external temperature, the burners of the hot water boilers come into operation, which at the first stage will provide an increase in the thermal agent; this will then be passed on to the supply of the thermal agent in nominal conditions, i.e., Td/tint = 105/70 °C.

### 4.1.2. Engine-Generator Groups

Each engine-generator assembly will contain:

- fuel supply system;
- water system (cooling water and process water system);
- engine block and coolers (water, oil);
- HV (high-voltage) recovery system;
- LV (low-voltage) recovery system;
- lubrication system;

- systems at the electric generator, regarding:
  - o  power and voltage regulation including unit synchronization;
  - o  alternator excitation adjustment;
  - o  vibration detection and monitoring;
  - o  gas exhaust by-pass automation;
  - o  starting system;
  - o  gas leak detection system.

The instrumentation is in accordance with the regulations in force and includes:

- pressure sensors and pressure switches;
- temperature sensors and thermostats;
- speed analyzers;
- level and flow micro-switches;

All sensors mounted on the motor generators will be identified by steel plates with their tags.

The steering of the engines will be achieved with an engine management system, which will be mounted on the boards related to the engines.

### 4.1.3. Engine Control Center (ECC)

The ECC engine control panels control the engine's start and stop functions, as well as other dedicated operations. In addition, the panels contain all the necessary equipment to supply and control the auxiliary elements of the engines, as well as to supply all the alarms and equipment states. The generator control panel includes all control, protection, and visualization elements.

The automation system of each engine will be connected to the MODBUS network using the SCADA system of the boiler. Thus, on the PC SCADA monitor, the main technological parameters of the motor will be visualized, as well as the main electrical parameters of the generator operation, such as equipment status, alarms, and events.

### 4.1.4. Hot-Water Boilers (HWB)

Hot-water boilers operate on both gas and oil fuel. They are equipped with a mixed gas-liquid burner, with a modulating adjustment mode for the gas fuel and four stages on the liquid fuel.

- Thermal power 3500 kW;
- Gas fuel consumption 125–350 $Nm^3$/h;
- Liquid fuel consumption 105–300 kg/.

The burner in this setup is made using a mono-bloc construction, with the possibility of folding on the left or right side, and has the following subassemblies:

- High-performance reverse tilt blower with low noise emissions;
- Suction circuit lined with soundproofing material;
- Air damper for air regulation, controlled by a stepper servo motor;
- Air-pressure switch;
- The fan started the motor at 2900 rpm, in a three-phase system, with zero at 50 Hz);
- A low emission combustion head, which can be adjusted according to the required power, equipped with a stainless steel combustion head, resistant to corrosion and of high-temperature steel;
- Maximum gas pressure switch, with a pressure test point, to stop the burner in case of overpressure in the fuel line;
- Flame control panel for system safety control including modulation controller;
- Ionization electrode for flame detection;
- Main terminal of power supply connections;
- Burner on/off switch;
- Auxiliary voltage LED signaling;

- Manual/automatic power increase/decrease button;
- LED signaling the burner operation;
- Contact motor and thermal relay with a release button;
- Internal engine thermal protection;
- Signaling engine failure by LED;
- LED signaling burner faults and an illuminated release button;
- LED signaling the correct direction of rotation of the fan motor;
- Panic button;
- Coded connection sockets;
- Degree of electrical protection (IP 54);
- T-shaped connector for DN 80 gas supply;
- Liquid fuel pump.

The gas ramp consists of a:

- Filter;
- Stabilizer;
- Minimum gas pressure switch;
- Safety valves;
- Leak test (for power > 1200 kW);
- One-stage operating valves, with an ignition gas flow regulator.

### 4.2. Automation Installations for Heating Boilers

Each boiler is equipped with an automation panel that manages all safety elements, temperature sensors, automatic valves, and boiler pumps.

#### 4.2.1. Internal Connections

There is an own-boiler temperature sensor. The panels are connected to each other and to the SCADA system on a MODBUS TCP/IP fieldbus. Thus, on the PC SCADA monitor, the main technological parameters of the boilers will be visualized, as well as the equipment status, alarms, and events.

#### 4.2.2. Automation/Installation of the Power Supply Circuit

This circuit is for the thermomechanical scheme of the plant, containing the automation elements that are necessary for the supply of hot water in the primary circuits of the campus, as well as for obtaining the necessary data for the preparation of thermal energy balances in automation equipment, such as:

- Instrumentation for monitoring the main parameters.
  - o Pressure transducers:
    - Pressure on the hot water circuits at the outlet and inlet to the heat recuperations of the thermal gas engines;
    - Pressure on water circuits, in addition to the motors and water, as well as the primary heating circuit;
    - Pressure on the drinking water-supply circuit of the water softening station;
    - Pressures on the primary heating circuit-flow (distributor) and return (pump discharge).
  - o Temperature transducers:
    - The temperature in the two areas (upper and lower) of the heat accumulators;
    - The temperature in the hot water circuit at the exit of the HWBs and when entering the equalization bottle;
    - The temperature in the hot water circuit at the outlet of each HWB;
    - The temperature in the water circuit from the storage tanks to each HWB;

■　　　The temperature in the equalization bottle.

- Instrumentation for measuring the thermal energy parameters:
  - o　Flow and return temperature measurement in the hot water circuits of the heat recuperations of the gas thermal engines.
  - o　Measuring the debit of the hot water on the flow (the output of the gas heat engines from the heat recuperation).
  - o　Measurement of the temperature on the flow and return of the hot water circuit's primary heating agent (input/output to the cogeneration plant).
  - o　Measuring the debit of the hot water on the flow leaving the cogeneration plant.
  - o　A thermal energy calculation element that will measure, calculate, and indicate the following parameters:
    - ■　Recovery flow temperature 1;
    - ■　Return temperature recuperation 1;
    - ■　Recovery flow rate 1;
    - ■　Instantaneous thermal power of the hot water supplied by the motor 1;
    - ■　Metering the thermal energy of the hot water supplied by the engine 1;
    - ■　Recovery flow temperature 2;
    - ■　Return temperature recuperation 2;
    - ■　Recovery flow rate 2;
    - ■　Instantaneous thermal power of the hot water supplied by the motor 2;
    - ■　Metering the thermal energy of the hot water supplied by the engine 2;
    - ■　Heating temperature of the cogeneration central outlet;
    - ■　Cogeneration central input recovery temperature;
    - ■　Cogeneration plant output flow rate;
    - ■　Instantaneous thermal power of the hot water provided by the cogeneration plant;
    - ■　Metering the thermal energy of the hot water provided by the cogeneration plant;
    - ■　Hot water temperature control at the output of the storage tanks-input to each HWB.

## 5. Implementation and Discussions

This paper presents the general architecture of a BMS system for energy consumption management within a university campus. In this section, we will present and discuss some of the basic elements regarding implementation; a future paper will present in detail the implementation method and the obtained results. The solution proposed in this paper aims to reduce the energy consumption by 13% for thermal energy and by 32% for electricity.

The proposed system uses the latest technology in the field of BMS systems. The automation of the cogeneration motors will be performed with a PLC, monitored by a SCADA system, interconnected throughout with the buildings via a fieldbus. A BMS system will be used to control the temperatures in each room, which can be used in all modern buildings.

The software architecture of the system is presented in Figure 5. Thus, the temperature probes and electro-valves in the students' houses are interconnected via the Modbus RTU fieldbus. In the control room at the students' house level, the BMS system is installed, which includes a driver for the MODBUS RTU. The connection to the fieldbus is made via a USB-RS485 interface. From the control room, you can view the temperatures in each room; you can also set the desired temperature in each room (for example, in unoccupied rooms, the temperature will be set to the minimum possible value to reduce unnecessary energy consumption).

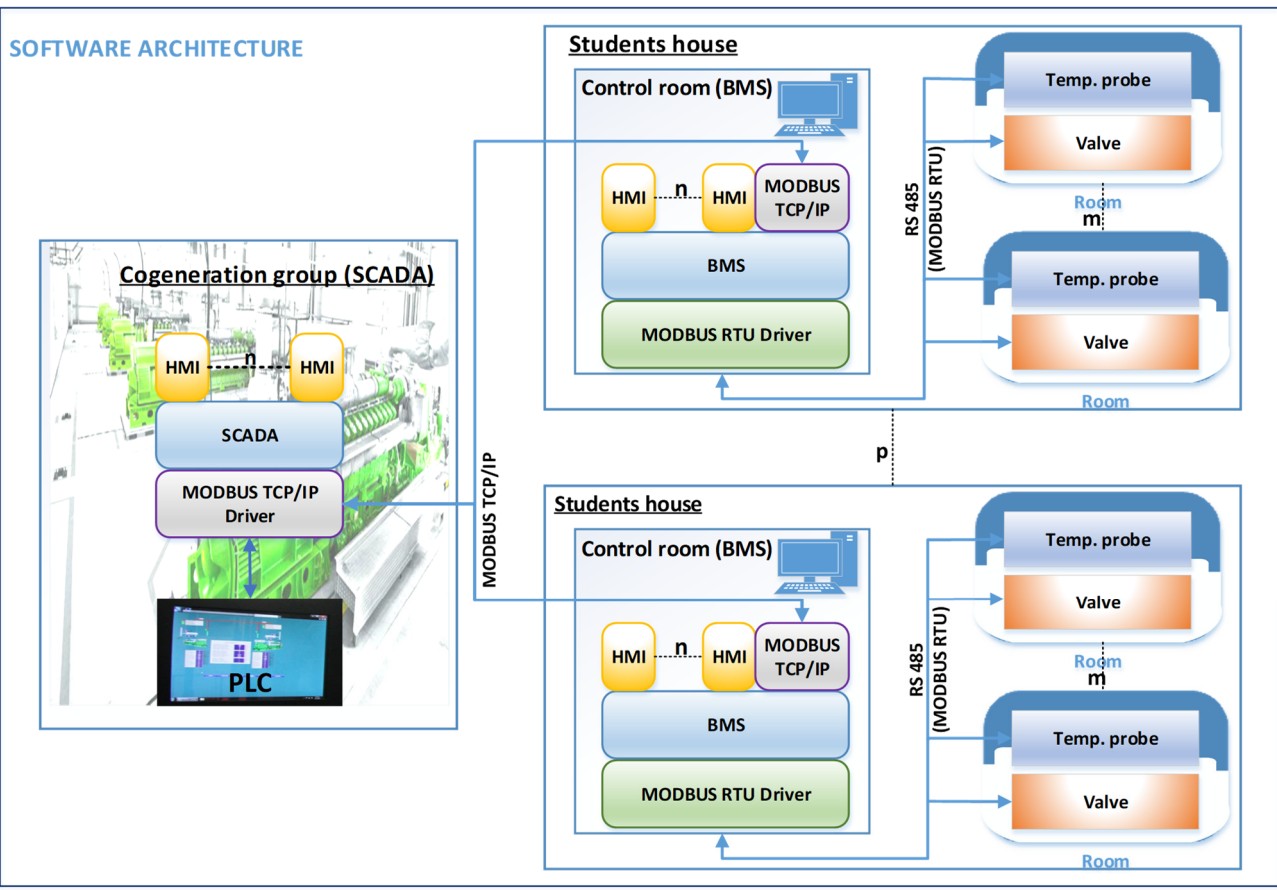

**Figure 5.** The software architecture of the proposed solution.

As can be seen from Figure 5, there may be several student houses ("p" in Figure 5) that are monitored, and each students' house has a varied number of rooms ("m" in Figure 5). All temperature control is performed at the building level via BMS. The acquisition of the values for the temperatures and the setting of the solenoid valves in each room are performed periodically, at 5 s intervals (this time period is enough because it is a slow process and this allows the monitoring of an increased number of rooms on the same fieldbus).

The BMS also has a Modbus TCP/IP module, through which the SCADA system from the cogeneration group will be able to determine the energy demand for that building. The BMS system will make the necessary predictions (based on the outside temperature and the difference in each room between its temperature and the temperature that must be maintained). This requirement is used by the SCADA system to control the PLC that controls the cogeneration unit. The SCADA system will connect to the BMS and PLC systems that control the cogeneration group via the MODBUS TCP/IP fieldbus. Through the SCADA system, different parameters can be viewed through more HMI interfaces (dashboards) ("n" from Figure 5), as will be seen below.

The BMS system developed in this case can be used simultaneously in numerous buildings. For this reason, the system is distributed between buildings, using a middleware system based on the TCP/IP stack (the buildings can be interconnected via a local network or through the public internet network). The modules in a building are connected to the BMS system via a local fieldbus (MODBUS TCP/IP).

Figure 6 presents some dashboards from the system control room. From here, the cogeneration plant and the temperature from each room can be controlled and monitored. A detailed dashboard for the control and monitoring of the cogeneration plant is shown in Figure 7. The temperature from each room can be controlled at the level of each building and from the control rooms of the system.

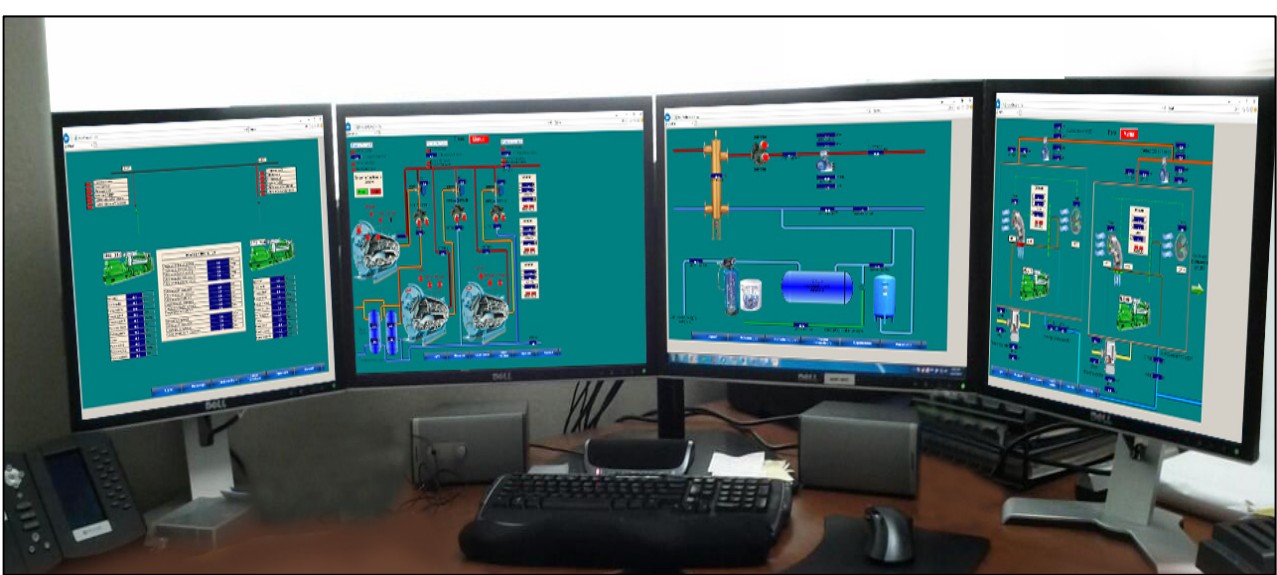

**Figure 6.** The dashboards for the students' house and cogeneration plant.

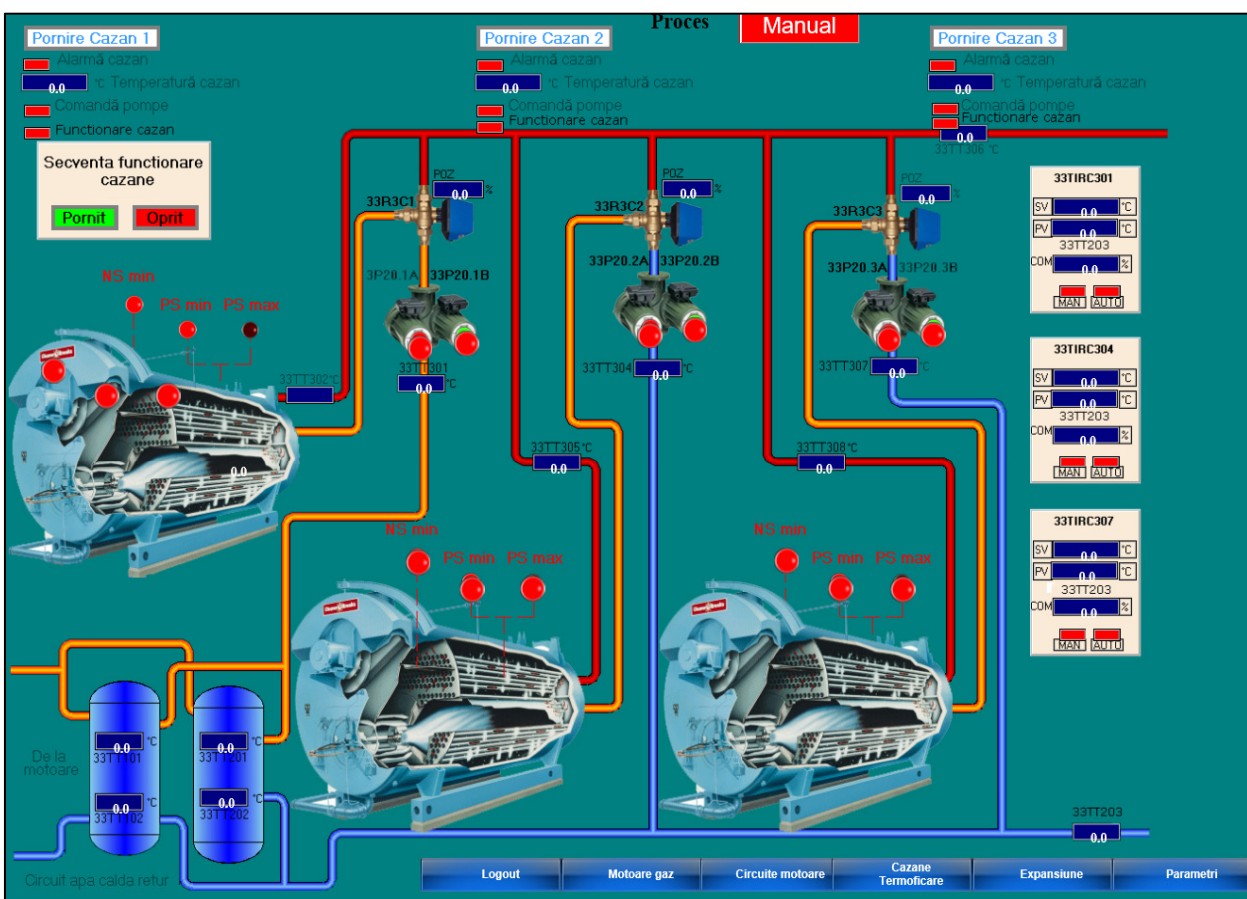

**Figure 7.** A detailed dashboard for the cogeneration plant.

Figure 8 presents the HMI interface of the PLC that monitors and controls the cogeneration plant. This PLC is connected to the SCADA system installed in the control room via a local fieldbus—the MODBUS TCP/IP.

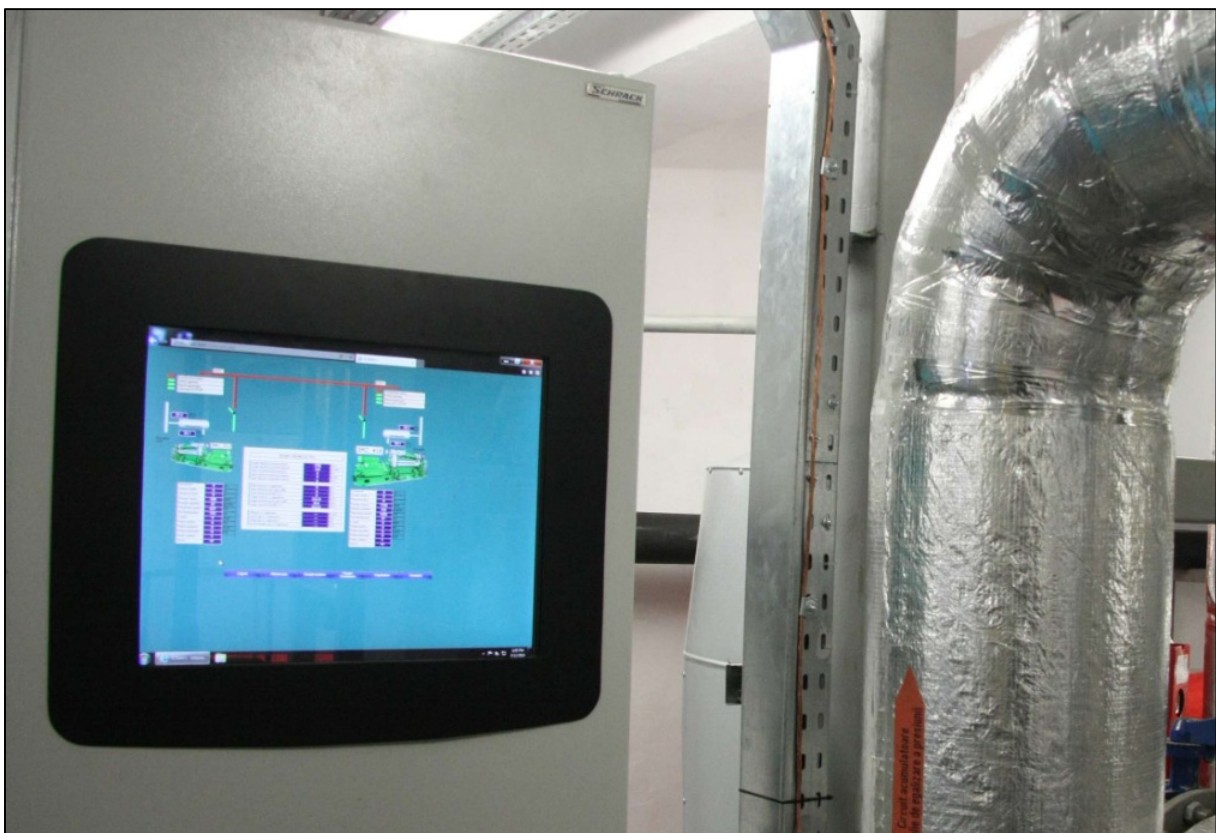

**Figure 8.** The PLC that controls and monitors the cogeneration plant.

The proposed solution has reduced the consumption of thermal energy by 13% and the consumption of electricity by 32%. At this moment, the system is functional; the problems that existed during the implementation were social because the tenants of the building had to be informed that the monitoring of energy consumption for each room was performed only to reduce the cost of energy consumed. We can say that the initial goal was achieved in terms of reducing energy consumption. We can still work on optimizing the system in the future to achieve an even greater reduction in consumption and expand its functionality during the hot season. Due to its specific nature (the system is developed for campus buildings that have their own specific setups in terms of construction and heating system), the system in the study cannot be compared with other systems.

## 6. Conclusions

The energy efficiency of buildings is becoming an increasingly important issue. Efficient energy management can lead to significant savings in terms of financial costs and energy consumption (which is especially important in the current context of global warming). This paper presents a solution that has been adopted for the energetic rehabilitation of a students' dormitory within a university campus. The solution involves installing a temperature sensor and an electro-valve in each room in order to control the room temperature, depending on the activity in the room, the expected weather conditions, and the desired room temperature. The adopted solution led to significant efficiency in the heat consumption achieved in the student dormitory. The main contribution of this paper is that the sensors and actuators installed in each room are included in a functional system for the management of energy consumption in the cold season. The system proposed in this paper reduced the consumption by 13% in terms of heating energy and by 32% in terms of electrical energy. In the future, it is planned that the system should be used in the hot season by installing air conditioning systems in each room that are controlled by the system.

In conclusion, we can say that the initial goal (reducing the consumption of thermal energy and electricity) has been achieved.

The system is currently implemented for the cold season, with the main purpose of thermal energy management, but in the future, it is planned to expand the system for the hot season by installing air conditioning systems in each room. To further reduce the consumption of electrical energy from the main grid, the system can be expanded by using renewable energy sources, such as solar panels. These renewable energy sources can lead to a decrease in electricity consumption in the centralized system, especially in the hot season.

**Author Contributions:** Conceptualization, N.C.G., I.U., C.R. and C.F; methodology, N.C.G., I.U., C.R. and C.F.; software, N.C.G. and I.U.; validation, C.R. and C.F.; formal analysis, N.C.G. and I.U.; investigation, N.C.G., I.U., C.R. and C.F.; resources, C.R. and C.F.; data curation, N.C.G., I.U., C.R. and C.F.; writing—original draft preparation, N.C.G. and I.U.; writing—review and editing, N.C.G., I.U., C.R. and C.F.; visualization, N.C.G. and I.U.; supervision, C.R.; project administration, C.F. All authors have read and agreed to the published version of the manuscript.

**Funding:** This research received no external funding.

**Informed Consent Statement:** Not applicable.

**Data Availability Statement:** Data sharing is not applicable.

**Conflicts of Interest:** The authors declare no conflict of interest.

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
