# Peer review of "An Optimizing Heat Consumption System Based on BMS"

_applsci, doi:10.3390/app12073271_

Round 1
Reviewer 1 Report
Dear Authors,
Thank You for sound analysing and detailing execution of state of the art optimising heat consumption with BMS systems for existing dormitories retrofit project!
Paper " An optimized heat consumption system based on BMS" shows up in first part with sound analysis of current state of the art sensor based heating energy efficiency strategies from a utilities planning perspective. In second part utilities hard- and software system compoments are intruduced, based on gas and oil plant energies used. In future paper part monitoring data shall be available.
Basic academic line for paper is definitely convincing, although some thoughts towards a more holistic overall approach would have been possible and of definite readers value , e.g.:
In regard of critical overheating and comfort in summer phase analysing cooling problem
In regard of existing dormitories building strucure implementing sensors in building shell, windows, for e.g. possible variable insulation shutters in crisp cold nights, as well as solar shading installation in summer phase
In regard of current only fossil plant resources gas and oil at least including in planning outlook future passive energies as e.g. PV systems
Reviewer 2 Report
1. The Abstract need observation and major results and incomes.
2. Avoid the long citations of [6 - 9], [10 - 14]….
3. Installing a temperature sensor and an electro-valve in each room to control the room temperature will enhance the efficiency, so what is the contribution of this paper?
4. There is no economic study.
5. In the title, there is an optimizing … but in the paper do you not using any optimization method or techniques.
6. You said “purpose of this article was to propose an architecture”, what is the relationship between architecture and optimization cited in the title?
7. How can I know that the proposed architecture is the best solution, there is no indices and no comparisons.
8. In the results, please cite some of your future work. I think the extension of this work is not a perspective.
Reviewer 3 Report
Comments and Suggestions for Authors
- Abstract
1.1. The authors should add information about the methodology and the main findings.
1.2. Lines 18-19 - "we propose an optimizing heat consumption system based on BMS (Building Management System) for the university campus energy.” - Please be more specific (See the comments about lines 44 - 45).
- Introduction
2.1. The authors should better highlight the innovation and relevance of their research.
2.2. Lines 44 - 45 - "In this paper, we propose a solution for a smart system for optimizing heat consumption developed to rehabilitate the students’ dormitories from a university campus." and Lines 48 to 50 - “The main objective of this paper is to propose an optimizing heat consumption system that ensures the medium and long term functionality of the accommodation service of a campus.” – The authors should more clearly define the objective and delimitation of the work. Also, is the system designed for all types of students’ dormitories?
- Body part of paper
3.1. Lines 241 – 242 “The purpose of this article is to present a system of automation - control and supervision for primary thermal and electrical equipment as well as auxiliary equipment.” - See the comments about lines 44 – 45.
3.2. Lines 444 to 447 “This paper presents the general architecture of a BMS system for energy consumption management within a university campus. In this section, we will present and discuss some basic elements regarding the implementation, following that another extended paper will be present in detail the implementation method and the obtained results” - This may be why the discussion of the results is weak and superficial. In a journal of the quality of Applied Science, the authors should have presented more robust results and discussions.
- Conclusions
4.1. The conclusion section should be completely reformulated. The authors should better highlight the relevance of the results, recommendations for future works, and limitations of the research.
4.2. Lines 513 to 515 - “The paper contains discussion about some basic elements for implementation, and in the future we will extend the present paper and we will present in more detail the implementation method and the obtained results.” – Please remove.

Round 2
Reviewer 2 Report
- Abstract: what mean by “de” in this sentence (to heat “de” building).
- In the sentence “to propose “an” control system”, is should be “a”.
- Please revise the English language of this paper.
- Remove the future work from the section of “Implementation and discussions”.
- To prove your system enhancement, I think is better to display the first consumption (before your work) and the reduced consumption (after your work), like this we can compare and prove that your work is effectively.
Reviewer 3 Report
Comments and Suggestions for Authors
Dear authors,
The discussion of the results remains weak. Section 5 just presents the system. Authors need to address the performance of the system once it is up and running. During the system deployment process, did everything go as expected? Is the system performance as expected? Have performance targets been met? There is need to properly situate your findings in the context of the findings of other researchers.

Round 3
Reviewer 3 Report
Reviewer Comments to Author
Dear authors,
Thank you for addressing my comments.
Author Response
THANK YOU for your kind response.